# The Lanthipeptide Synthetase-like Protein CA_C0082 Is an Effector of Agr Quorum Sensing in *Clostridium acetobutylicum*

**DOI:** 10.3390/microorganisms11061460

**Published:** 2023-05-31

**Authors:** Jonathan R. Humphreys, Zak Bean, Jamie Twycross, Klaus Winzer

**Affiliations:** 1BBSRC/EPSRC Synthetic Biology Research Centre (SBRC), School of Life Sciences, University Park, The University of Nottingham, Nottingham NG7 2RD, UK; jonathan.humphreys@nrel.gov (J.R.H.); zak.bean@chainbiotech.com (Z.B.); 2School of Computer Science, Jubilee Campus, The University of Nottingham, Nottingham NG8 1BB, UK; jamie.twycross@nottingham.ac.uk

**Keywords:** quorum sensing, Agr system, clostridium acetobutylicum, ABE fermentation, sporulation, lanthipeptide synthteases, LanM, granulose

## Abstract

Lanthipeptide synthetases are present in all domains of life. They catalyze a crucial step during lanthipeptide biosynthesis by introducing thioether linkages during posttranslational peptide modification. Lanthipeptides have a wide range of functions, including antimicrobial and morphogenetic activities. Intriguingly, several *Clostridium* species contain lanthipeptide synthetase-like genes of the class II (*lanM*) family but lack other components of the lanthipeptide biosynthetic machinery. In all instances, these genes are located immediately downstream of putative *agr* quorum sensing operons. The physiological role and mode of action of the encoded LanM-like proteins remain uncertain as they lack conserved catalytic residues. Here we show for the industrial organism *Clostridium acetobutylicum* that the LanM-like protein CA_C0082 is not required for the production of active AgrD-derived signaling peptide but nevertheless acts as an effector of Agr quorum sensing. Expression of CA_C0082 was shown to be controlled by the Agr system and is a prerequisite for granulose (storage polymer) formation. The accumulation of granulose, in turn, was shown to be required for maximal spore formation but also to reduce early solvent formation. CA_C0082 and its putative homologs appear to be closely associated with Agr systems predicted to employ signaling peptides with six-membered ring structures and may represent a new subfamily of LanM-like proteins. This is the first time their contribution to bacterial Agr signaling has been described.

## 1. Introduction

*Clostridium acetobutylicum* is a Gram-positive, strictly anaerobic, spore-forming bacterium well known for its ability to convert starches and monomeric sugars into organic acids and solvents, including acetone, butanol, and ethanol [1]. Used in the first decades of the twentieth century in large-scale industrial fermentations, it now serves as a model organism for the study of solvent metabolism and the process of clostridial endospore formation [2,3].

In a typical batch culture, sugars such as glucose are first converted to acetic and butyrate acids, together with copious amounts of CO_2_ and H_2_. Following acidification of the culture medium, cells cease to divide and transition to the stationary phase. At this stage, some of the produced acids are taken up again and converted to solvents, along with the remaining sugar. In addition to this metabolic shift to solvent formation, cells also start to accumulate a glycogen-like storage compound termed granulose before eventually initiating the irreversible process of endospore formation [1,4,5]. The role of granulose is poorly understood, but it is believed to provide carbon and energy for this last stage of the clostridial life cycle [4].

Despite recent progress, we only have a limited understanding of how the metabolic shift to solvent formation and the initiation of sporulation are regulated. It is well established, however, that the master regulator Spo0A plays a key role in both processes as it is required to produce large amounts of solvents and is absolutely essential for endospore formation [6,7]. Activation of Spo0A is achieved by phosphorylation, which, in contrast to *Bacillus* spp., is carried out directly by orphan histidine kinases, presumably in response to yet unknown environmental and intracellular signals [8].

Recently, we have started to investigate the contribution of social signals, such as those generated by the *agr* and RNPP-type quorum sensing systems present in the organism [9,10,11]. While *agr* quorum sensing systems are known to be widespread in the class *Clostridia* [12], comparatively little is known about their role in these organisms, particularly in non-pathogenic species. It was recently shown that inactivation of the *agr* system in *C. acetobutylicum* reduced endospore formation and abolished granulose accumulation while having no obvious effect on the final concentrations of acids and solvents produced [10].

Agr quorum sensing systems typically consist of four genes, *agrBDCA*, which are often arranged in a single operon [13,14]. *agrD* encodes the signaling peptide precursor, which is processed and exported by the membrane-associated AgrB protein. Following accumulation of the mature, cyclic signaling peptide in the environment, it binds to the AgrC histidine kinase, which in turn leads to phosphorylation and thus activation of the AgrA response regulator. In *C. acetobutylicum*, unlike *Staphylococcus aureus*, the four genes appear to be organized in two operons, *agrBD* and *agrCA*. An associated regulatory RNA mediating Agr-dependent gene activation, as found in *S. aureus*, appears to be missing. Interestingly, the gene located immediately downstream of the *agrCA* operon in *C. acetobutylicum*, CA_C0082, is annotated to encode a ‘lanthipeptide modifying enzyme’ (NCBI accession number WP_010963409.1) and reported to possess similarity to lanthipeptide synthetases [15]. In lanthipeptide-producing organisms, these enzymes are required for the introduction of intramolecular bonds within the maturing lanthipeptide precursor [16]. Given that *C. acetobutylicum* is devoid of all other genes required for lanthipeptide synthesis, the location of such a gene next to the *agr* locus was intriguing and suggested a possible functional relationship.

The above findings motivated the study of CA_C0082 in the context of Agr-dependent quorum sensing. More specifically, the aim of the work presented here was to establish whether the CA_C0082 gene product was required for the generation of the Agr signaling peptide or contributed in other ways to the expression of Agr-dependent phenotypes.

## 2. Methods

### 2.1. Bacterial Strains and Media

Bacterial strains utilized in this study are listed in Table 1. *C. acetobutylicum* ATCC 824 and generated mutants were grown at 37 °C in an anaerobic cabinet (MG1000 Anaerobic Work Station, Don Whitley Scientific, Bingley, UK) containing an atmosphere of 80% nitrogen, 10% hydrogen, and 10% carbon dioxide. The organism was routinely cultured in supplemented clostridial basal medium (CBMS), unless stated otherwise. CBMS was based on CBM as previously described [17], but contained glucose (50 g/L unless otherwise stated) and calcium carbonate (5 g/L) as buffering agents. For agar plates, 15 g/L agar was added, while calcium carbonate was omitted. *Escherichia coli* TOP10 was grown in lysogeny broth (LB) at 37 °C. Antibiotics were used at the following concentrations: chloramphenicol, 25 μg/mL; erythromycin, 40 μg/mL; tetracycline, 10 μg/mL; and thiamphenicol, 15 μg/mL. *C. acetobutylicum* wild type and all mutants generated in this study were stored as spore stocks.

### 2.2. Plasmids, Primers, DNA Techniques

Plasmids used in this study are listed in Table 2. Primers are listed in Appendix A and were synthesized by Eurofins MWG Operon, Ebersberg, Germany. PCR amplifications were carried out using high-fidelity Phusion polymerase or *Taq* DNA polymerase (both from New England Biolabs, Hitchin, UK). Electroporation of *C. acetobutylicum* was performed as described previously [20]. Plasmid isolation and genomic DNA preparations were carried out using the QIA-prep Miniprep kit (Qiagen, Manchester, UK) and DNeasy Blood & Tissue kit (Qiagen, Manchester, UK), respectively. Restriction enzymes were supplied by New England Biolabs and Promega and were used according to the manufacturers’ instructions.

### 2.3. Construction of Mutants Using Clostron Technology

*C. acetobutylicum* ATCC 824 ClosTron mutagenesis was carried out as described by Heap et al. [18,21]. ClosTron-carrying plasmids were designed using the design tool available at http://www.ClosTron.com/ClosTron2.php (last accessed on 12 June 2021) and purchased from ATUM (formerly DNA 2.0). Numbers in the respective plasmid names (Table 2) indicate the retargeting site used within the disrupted gene. Genomic DNA from putative mutants was screened by PCR to establish whether the ClosTron-derived group II intron had been inserted at the desired site and included (i) primers that annealed on either side of the target site (Ca_c0082 -sF1/Ca_c0082 -sR1) and (ii) individual flanking primers together with a group II intron-specific primer, EBS universal (thus amplifying the intron–exon junctions). The generated PCR fragments were sequenced to obtain definite proof that intron insertion had occurred at the desired position. Single intron insertion as well as insertions sites were further confirmed by Illumina whole genome sequencing carried out by a third-party provider (Microbes NG, Birmingham, UK).

### 2.4. Generation of Complementation Vectors

To construct the CA_C0082 complementation vector pMTL85141-CA_C0082, a 2354 bp fragment containing the CA_C0082 gene and a 294 bp 5′ non-coding region expected to contain the gene’s native promoter were amplified by PCR from genomic *C. acetobutylicum* ATCC 824 DNA using the primer pair Ca_c0082-F1/Ca_c0082-R1 (Appendix A). These contained NotI and NheI restriction sites, respectively, so that the resulting fragment could be cloned into the equally digested clostridial shuttle vector pMTL85141 [20]. The resulting vector was confirmed by restriction analysis and Sanger sequencing. Constitutive expression of CA_C0082 was achieved through Hi-Fi assembly (New England Bioscience, NEBuilder^®^ HiFi DNA Assembly Master Mix, Hitchin, UK) of the PCR-amplified CA_C0082 structural gene, using primer pair Ca_c0082-F2/Ca_c0082-R2, and plasmid pMTL85143 [11], containing the *C. sporogenes fdx* promoter, linearized through digestion with NdeI and NheI.

### 2.5. Generation of a GusA Reporter Plasmid

Plasmid pMTL-JL1 (Jessica Locker, University of Nottingham, unpublished), created as a test vector for the expression of *Streptococcus agalactiae gusA* [22], was digested with NotI and EcoRI to remove the *fdx* promoter region but maintain the *fdx* ribosome binding site (RBS) and *gusA* structural gene. The 88-bp intergenic region between *agrCA* and CA_C0082 was amplified by PCR using primers P_Ca_c0082-F/P_Ca_c0082-R, digested with NotI and EcoRI, and cloned into the above vector upstream of *gusA*. The resulting CA_C0082-*gusA* reporter construct was confirmed by sequencing and named pMTL-JL1-P_CA_C0082_.

### 2.6. Spore Assays and Detection of Granulose

*C. acetobutylicum* strains were grown in either 5 mL or 30 mL CBMS, depending on the experiment, to enable sporulation. 200 μL samples of culture were taken at the indicated time points and incubated at 80 °C for 10 min. Serial dilutions were carried out, and 10 μL aliquots of the heat-treated cell suspension were plated onto CBM agar. Agar plates were incubated for 48 h before CFUs were enumerated. For each assay, a *spo0A* mutant [19] and the wild type were included as negative and positive controls, respectively. Heat-resistant CFUs determined for the wild type varied according to culture conditions, including culture volume, and reached approximately 2 × 10^5^ per mL and 5 × 10^7^ per ml for 30 mL and 5 mL cultures, respectively.

To assess the visual accumulation of granulose, *C. acetobutylicum* strains were grown as cell lawns on CBM agar containing 5% glucose. Cell lawns formed after adding 10 μL aliquots of cell suspension onto the agar surface and incubating at 37 °C for 48 h. These were stained with iodine, as previously described [10]. For the quantitative granulose assay, *C. acetobutylicum* strains were grown in 10 mL CBMS for 48 h. At this point, 1 mL samples were taken and centrifuged at 16,000× *g* for 2 min, and the supernatant was removed. Cell pellets were placed on ice and washed 3 times using 1 mL ice-cold PBS, centrifuging at 16,000× *g* for 2 min after each wash. The final pellet was resuspended in 300 μL ice-cold PBS and kept on ice. Cells were then sonicated using the Biorupter^®^ Plus (Diagenode, BE, USA) for 20 cycles of 30 s pulses to release cell-based granulose. Protein concentration was subsequently determined using Pierce^TM^ BCA Protein Assay Kit (Thermo Fisher Scientific, Loughborough, UK), following the manufacturer’s instructions. After protein determination, samples were centrifuged at 16,000× *g* for 2 min, and the supernatant was removed. Granulose within the cell pellet was subsequently determined using a Glycogen Assay Kit (Sigma-Aldrich, St Louis, MO, USA), following the manufacture’s instructions. Granulose was displayed as the μg of granulose per μg of protein found within each sample.

### 2.7. Addition of Synthetic AgrD Signaling Peptide

Synthetic *C. acetobutylicum* AgrD peptide (R6T0) was synthesized and purified by Peptide Protein Research Ltd. (Fareham, UK). The lyophilized peptide was dissolved in DMSO to obtain a 10 mM stock solution and used as previously described [10].

### 2.8. Analysis of Fermentation Products

*C. acetobutylicum* ATCC 824 wild type and mutant derivatives were grown in 50-mL Falcon tubes containing 30 mL of CBMS. At the indicated time points, 1 mL samples were removed, placed on ice, and centrifuged at 16,000× *g* for 5 min to obtain cell-free culture supernatant. Extraction of fermentation products and their gas chromatographic analysis were carried out as described previously [10].

### 2.9. GusA Reporter Assay

*C. acetobutylicum* ATCC 824 pMTL-JL1-P_CA_C0082_, *C. acetobutylicum agrB::*CT*ermB* pMTL-JL1-P_CA_C0082_, and *C. acetobutylicum agrA::*CT*ermB* pMTL-JL1-P_CA_C0082_ mutants were grown in 50-mL Falcon tubes containing 30 mL of CBMS. At indicated time points, 1 mL samples were removed, OD_600_ recorded, centrifuged at 21,000× *g* for 2 min, and supernatant removed to provide a cell pellet. Pellets were stored at −20 °C until used in the assay. Pellets were resuspended in 500 μL of buffer consisting of 60 mM Na_2_HPO_4_, 45 mM NaH_2_PO_4_, 10 mM KCl, 1 mM MgSO_2_ ⋅ 7 H_2_O, 50 mM β-mercaptoethanol pH 8.5, and 75 μL of the resuspension inoculated into a 96-well plate before the addition of 25 μL of buffer containing 40 μg/mL 4-methylumbelliferyl-β-D-glucuronide. Following excitation at 355-375 nm, fluorescence was measured between 440 and 460 nm for 30 min. Change in fluorescence across the linear range for each sample was determined and normalized against the OD_600_.

### 2.10. Phylogenetic Analysis

Protein amino acid sequences were aligned using MUSCLE v3.8.31 [23] with default settings, and resulting duplicate alignments were removed. Best-fit models for amino acid replacement were then selected using ProtTest v3.4.2 [24]. ProtTest reported the LG + I + G + F model as the best-fit model based on both AIC and BIC measures. This model was used in phylogenetic inference. Bayesian MCMC inference was performed using MrBayes v3.2.5 [25]. MrBayes was run with two independent analyses of eight chains (one cold, seven heated), which were sampled every 100 generations. Runs were deemed to have converged when the average standard deviation of split frequencies dropped below 0.01, using 25% burn-in. After runs had converged, a 50% majority rule consensus tree was computed using the final 75% of runs. Maximum-likelihood inference was performed using RaxML v8.2.10 [26]. The autoMRE bootstrap criteria were used to determine the number of bootstrap replicates to run. Using the 102 bootstrapped trees generated, a 50% majority rule consensus tree was computed using the SumTrees program, part of the DendroPy package [27].

### 2.11. Statistical Analysis

All data are presented as mean ± standard deviation (SD) for three independent replicas. Statistical analyses were performed using GraphPad Prism software version 9 (GraphPad, La Jolla, CA, USA), employing a paired *t*-test for side-by-side comparisons of the wild type and a given mutant under a particular condition, or for side-by-side comparisons of the same strain under two different conditions, with *p* values below 0.05 considered to be statistically significant.

## 3. Results

### 3.1. CA_C0082 and Its Close Homologs Lack a Zinc Binding Motif and Are Associated with Agr Quorum Sensing

In *C. acetobutylicum*, CA_C0082 is located 88 bp downstream of the *agrA* response regulator gene and annotated to encode a protein with similarity to lanthipeptide synthetases. To obtain clues towards its biological role, the presence of close homologs in other bacterial species and their genetic organization were investigated. BlastP searches of available bacterial genomes revealed putative CA_C0082 homologs to be present in strains of eleven clostridial species, all of which are non-pathogenic and known to produce short-chain fatty acids and solvents such as ethanol or butanol (see Appendix A for a complete list of species, strain names, and identified sequences). In all but one species, the identified homologs are located immediately downstream of a complete *agr* gene cluster consisting of separate *agrBD* and *agrCA* operons, as in *C. acetobutylicum* (Figure 1A). Interestingly, the *agrD* genes of these systems are all predicted to encode autoinducing peptides (AIPs), which in their final, mature form consist of 6-membered ring structures rather than the far more common five-membered thiolactone rings (Appendix A and Appendix A). It was therefore hypothesized that the CA_C082 protein may play a role in AIP modification.

However, sequence comparison of the different CA_C0082 protein homologs with established lanthipeptide synthetases revealed that their cyclase domains lacked conserved motives and amino acid residues required for binding of an active site zinc ion (Appendix A) [28]. Similarly, conserved residues in the dehydratase domain with established roles in the phosphorylation and elimination steps were also lacking (Appendix A). Indeed, while the CA_C0082 protein and its homologs were found to be most similar to LanM-type lanthipeptide synthetases, they were somewhat smaller, consisting of 670 to 728 amino acids rather than the typically observed 900–1200 residues [29], mainly due to the absence of parts of the dehydratase domain (Appendix A).

Phylogenetic analyses were undertaken, as described in the Methods, using both Bayesian Markov chain Monte Carlo (MCMC) and maximum-likelihood inferences [25,26]. For this, the identified CA_C0082 homologs were compared with selected LanM proteins representing (i) confirmed LanM-type synthetases (LanM2 and LctM from *Bacillus licheniformis* and *Lactococcus lactis*, respectively), (ii) putative LanM homologs to which CA_C0082 and related proteins were most similar, and (iii) putative full-length LanM homologs that were also present in the genomes of strains containing a CA_C0082 homolog. Overall, Bayesian and maximum-likelihood inference produced very similar phylogenetic trees (Figure 1B and Appendix A). CA_C0082 homologs formed a distinct subcluster, separate from the putative full-length LanM synthetases with which they shared the highest similarity. They also had no particularly close association with other putative full-length LanM enzymes present in some of the organisms harboring CA_C0082 homologs. It was therefore interesting that some of the identified CA_C0082 homologs and associated *agr* quorum sensing systems formed part of lanthipeptide biosynthesis gene clusters, e.g., in *Clostridium hungatei*, while no such genes were present in the *C. acetobutylicum* genome (Figure 1A).

### 3.2. Generation and Characterization of CA_C0082 Mutants

Given the lack of a zinc binding motif and the absence of other known lanthipeptide biosynthesis genes in the *C. acetobutylicum* genome, it was possible that CA_C0082 was a degenerate, non-functional remnant of an ancestral lanthipeptide synthetase gene or that the encoded protein had evolved to take on a different role no longer linked to peptide modification. The encoding gene was therefore inactivated through ClosTron mutagenesis [18] to identify phenotypes potentially linked to CA_C0082 function. Two different mutants were made using ClosTron plasmids pMTL007C-E2::CA_C0082-1116|1117S and pMTL007C-E2::CA_C0082-538|539A (see Section 2.3 and Table 2), resulting in the insertion of the ClosTron construct in sense and antisense orientation between base pairs 1116/1117 and 538/539 of the CA_C0082 coding sequence, respectively. Successful integration of the respective ClosTrons at the anticipated sites was confirmed for both mutants through PCR of the CA_C0082 encoding region and Sanger sequencing of the resulting PCR products. In addition, Illumina-based whole genome sequencing was carried out to independently confirm integration at the correct site and also to ensure the presence of a single insertion for each mutant, i.e., the absence of additional ClosTron insertions in other parts of the genome. Two independent mutants, both with only a single intron insertion at the CA_C0082 gene confirmed by whole genome sequencing, were obtained and referred to as *C. acetobutylicum* CA_C0082::CT*ermB*_s and *C. acetobutylicum* CA_C0082::CT*ermB*_as for the sense and antisense orientations, respectively. The inactivation of CA_C0082 resulted in a loss of granulose accumulation and a decrease in the number of heat-resistant colony-forming units (CFUs), here interpreted to represent the number of viable endospores, by one order of magnitude after 120 h of growth in CBMS containing 5% (*w*/*v*) glucose for both the CA_C0082::CT*ermB*_s and CA_C0082::CT*ermB*_as mutants (Figure 2). This matched the phenotypes previously reported for *agr* mutants [10], which were confirmed here in control experiments carried out with *agrB* and *agrC* mutants (Figure 2D and Figure 3C,D).

To confirm that the observed phenotypes were not caused by polar effects on the upstream *agrCA* operon or independent second site mutations, genetic complementation of the CA_C0082 defect was performed in strain *C. acetobutylicum* CA_C0082::CT*ermB*_s. Restoration of the granulose and sporulation phenotypes was observed upon complementation of the strain with plasmid pMTL85141-CA_C0082 expressing CA_C0082 under control of its native promoter (Figure 2). Iodine staining was initially used as a visual marker for gene complementation (Figure 2A). Strains able to produce granulose will stain blue/black (as seen for the wild type and complemented CA_C0082), unlike granulose-deficient strains, which will remain light orange in color (CA_C0082 mutant). Further confirmation of both granulose deficiency and complementation was seen with the use of a quantitative granulose assay (Figure 2B,C). As granulose is similar to glycogen, consisting of glucose residues linked via α-1,4- and (at branch points) α-1,6-glycosidic bonds, the use of a glycogen assay kit (Sigma-Aldrich, St Louis, MO, USA) was employed to establish a measurable quantity of granulose produced per µg of protein for each tested strain. As a granulose negative control, an existing granulose synthase (*glgA*) in-frame deletion mutant was used, which is unable to carry out the last step in granulose biosynthesis [19]. With this quantitative assay, mutants that stained granulose negative (*agrB,* CA_C0082, and *glgA*) were confirmed to contain no or very little granulose (<0.1 μg/μg protein) after 48 h of growth in CBMS containing 5% (*w*/*v*) glucose (Figure 2B). The wild type and complemented CA_C0082 mutant strains, however, produced significantly higher quantities of granulose in the same growth period (about 0.35 μg/μg protein) and were not statistically different from each other (Figure 2B,C). Spore production of the complemented strain matched the levels observed for the wild type, suggesting the absence of any significant polar effects upon *agrCA* expression. This was further confirmed through the introduction of plasmid pMTL85141-*agrCA* [10] into *C. acetobutylicum* CA_C0082::CT*ermB_*s*,* which failed to significantly alter its reduced sporulation phenotype. As the CA_C0082 gene is located directly downstream of the *agr* quorum sensing system, it was therefore determined that the loss of granulose and decrease in sporulation were caused by a loss of CA_C0082 functionality and not the result of an indirect impact upon the expression of the upstream *agr* quorum sensing system through polar intron insertion effects.

### 3.3. CA_C0082 Is Not Required for Active AIP Synthesis

Given its similarity to lanthipeptide synthetases, an initial hypothesis was that the CA_C0082 protein may contribute to the processing or modification of AgrD-derived cyclic AIP prior to its recognition by the AgrCA two-component system. The ability of the CA_C0082 mutants to generate biologically active AIP was therefore investigated by cross-streaking against the AIP-negative *agrB* mutant strain on CBM agar plates, as previously described [10]. Figure 3A shows that granulose production was restored in the *agrB* mutant streaks when located close to those of the CA_C0082::CT*ermB*_s strain, suggesting that the latter still produced the active peptide. However, it was also possible that the CA_C0082 mutant produced an inactive, diffusible AIP precursor that was converted to the mature, active form by the CA_C0082-positive *agrB* mutant cells. This would explain why the CA_C0082 mutants themselves remained granulose-negative. To investigate this possibility, the CA_C0082 mutants were also streaked against the wild type to provide a source of native AIP. This, however, also failed to induce granulose production in the CA_C0082 mutants (Figure 3B). Furthermore, the addition of 20 μM synthetic AIP, previously shown to effectively restore granulose and sporulation phenotypes in an *agrB* mutant [10], failed to induce sporulation in the CA_C0082 mutant, despite resulting in over 20-fold increases in final spore levels when added to the *agrB* mutant control (Figure 3D).

Taken together, the above experiments suggested that the CA_C0082 protein was not required for signal peptide processing and maturation. Rather, the generated CA_C0082 mutants appeared to be unable to initiate granulose production in response to the AIP signal.

### 3.4. CA_C0082 Expression Is Controlled by Agr Quorum Sensing

An alternative hypothesis was that following AIP accumulation and activation of the AgrCA two-component system, the CA_C0082 protein is in some way required for mediating the Agr response. This was tested by expressing the gene in a granulose-negative *agrA* response regulator mutant. Indeed, when plasmid pMTL85143-CA_C0082 was introduced into this background, expressing CA_C0082 from a constitutive *fdx* promoter, the resulting strain accumulated granulose, in contrast to the vector control (Figure 4A). Thus, expression of CA_C0082 was sufficient to restore granulose accumulation, and the biological activity of the encoded protein was independent of a functional Agr system. This explained why the CA_C0082 mutants displayed an *agr* mutant phenotype despite producing biologically active AIP and suggested that CA_C0082 expression may be controlled, directly or indirectly, via the AgrA response regulator.

β-glucuronidase (GusA)-based reporter assays were conducted to investigate the postulated regulation of CA_C0082 by Agr quorum sensing. Plasmid pMTL-JL1-P_CA_C0082_ was constructed that contained the *Streptococcus agalactiae gusA* gene [22] fused to the *agrCA-*CA_0082 intergenic region together with a non-native *fdx* ribosome binding site. The relative expression of this transcriptional *gusA* reporter fusion was determined in the wild type as well as in *agrA* and *agrB* mutant backgrounds by measuring GusA activity after 12, 24, and 48 h of growth. As can be seen from Figure 4B, the highest GusA activities were observed for the wild type at the 12 h time point, corresponding to the organisms’ entry into the stationary phase [10]. When expressed in *agrA* and *agrB* mutants, GusA activity was significantly reduced for the 12 and 24 h time points compared to the wild type (about 4–6 fold, *p* < 0.001), supporting the hypothesis that Agr quorum sensing controls expression of CA_C0082.

Furthermore, growth of the *agrB* pMTL-JL1-P_CA_C0082_ reporter strain in the presence of 20 μM synthetic AIP resulted in a 6-fold increase in expression over the DMSO control (Figure 4C), whereas in the *agrA* mutant, the reporter was unable to respond to the provision of synthetic AIP. Together, these findings confirmed the importance of a fully functioning Agr system for CA_C0082 expression.

### 3.5. Lack of Granulose Replicates the Sporulation Phenotype of Agr and CA_C0082 Mutants

Granulose storage has long been hypothesized to ensure the availability of carbon and energy in the later phases of sporulation [4]. It was therefore investigated whether loss of granulose availability had been an indirect cause of decreased sporulation for CA_C0082 and *agr* mutants. A previously generated *glgA* (granulose synthase) mutant incapable of synthesizing granulose [19] was employed for this purpose and served as a comparison.

It was hypothesized that a lack of granulose storage may have less severe effects under the commonly employed conditions of high glucose availability, and its impact may therefore have gone unnoticed in previous studies. To address this point, CA_C0082, *agrC,* and *glgA* mutants were grown in CBMS supplemented with 1–5% (*w*/*v*) glucose, with sporulation levels determined after 48 h and 120 h. Variation in glucose within this concentration range did not appear to markedly influence growth, as all strains were able to reach similar maximal ODs when entering the stationary phase, ranging from 2.1 to 2.8 across all tested strains and conditions (Appendix A). When grown in the presence of 4% and 5% (*w*/*v*) glucose, the granulose-deficient *glgA*, *agrA,* and CA_C0082 mutant strains all showed a similar, approximately 10-fold, decrease in final spore levels when compared to the wild type after 120 h (Figure 5B). A reduced sporulation in comparison to the wild type was also observed for all other glucose concentrations tested, but an interesting pattern emerged with respect to the endospore numbers formed by each strain in relation to the available glucose: spore levels for the wild type remained consistently high for glucose concentrations ranging from 2% to 5% (*w*/*v*), only showing a significant reduction when it was lowered to 1% (*w*/*v*). For the mutants, however, a reduction in spore numbers was already recorded for growth on 2% (*w*/*v*) and sometimes 3% (*w*/*v*) glucose, varying somewhat by mutant background and sampling time point. As a result, the differences observed when comparing the endospore levels reached with 5% and 1% (*w*/*v*) glucose, respectively, were more pronounced for the mutants, resulting in a 100-fold or greater decrease in spores for all granulose-deficient strains at the 120 h time point, whereas only a 20–30 fold reduction was observed for the wild type. Similar trends were observed for the 48 h time point. The overall sporulation pattern observed for the *glgA* mutant was similar to that of the *agrC* and CA_C0082 mutants, supporting the hypothesis that a lack of granulose was at least in part responsible for the reduction in sporulation for all three mutants.

### 3.6. Increased Early Formation of Solvents by CA_C0082 and Agr Mutants Is Linked to the Absence of Granulose

Given the effect of glucose at the above concentrations on spore formation, the respective mutant and wild type cultures were also analyzed for solvent production. Previous analysis of *agr* mutant strains had found no significant differences in solvent formation at 120 h compared to the wild type when grown in the presence of 5% glucose [10]. This was also seen here for all granulose-deficient mutants at the 120 h time point, with all strains showing regular increases in both total acetone and butanol as the initial glucose concentration was raised from 1% to 5% (Figure 6), in agreement with published data [4]. However, for the 48 h samples, a striking difference was observed. At this time point, all wild type cultures, when grown in the presence of 2% or greater glucose, showed similar acetone and butanol levels of about 20 mM and 30 mM, respectively, irrespective of the amount of glucose added. In contrast, the granulose-deficient strains showed incremental increases in 48 h acetone and butanol concentrations as the initial glucose concentration increased from 2% to 5%. As a result, at an initial glucose concentration of 5%, all granulose-negative mutants had produced significantly more solvents after 48 h than the wild type (*p* ≤ 0.05; Figure 6). When grown at lower glucose concentrations, the CA_C0082, *agrC*, and *glgA* mutants produced the majority of the total solvents already within the first 48 h of growth, with some examples resulting in a net loss due to evaporation by the 120 h time point.

The overall similar solvent profiles for *glgA*, *agrC,* and CA_C0082 mutants after 48 h suggested that a lack of granulose production was in some way responsible for the increased production of acetone and butanol.

## 4. Discussion

In this study, we investigated the role of the *C. acetobutylicum* CA_C0082 locus, predicted to encode an unusual LanM-like protein. CA_C0082 expression was shown to be regulated by Agr quorum sensing and required for granulose synthesis via a yet unknown mechanism. The ability to accumulate granulose, in turn, was shown to be required for optimal spore production and to affect early solvent formation.

The precise mechanism by which the CA_C0082 protein controls granulose formation remains to be established. Current evidence suggests that CA_C0082 homologs evolved from LanM enzymes, as they form a distinct subcluster among full-length LanM homologs (Figure 1) based on phylogenetic analyses including both Bayesian MCMC and maximum likelihood inferences. LanM-type (class II) lanthipeptide synthetases are bifunctional and consist of two domains: an N-terminal dehydration domain, which generates dehydroamino acids, and a C-terminal cyclization domain. The latter catalyzes the intramolecular addition of a cysteine-derived sulfhydryl group to the created dehydroamino acid, resulting in a ring structure [29]. Given the close association of CA_C0082 homologs with Agr systems predicted to employ AIPs with six-membered ring structures, formed via cyclization involving a cysteine residue, a role for CA_C0082 in AIP synthesis was initially postulated. However, this hypothesis was found to be incorrect, as CA_C0082 mutants still produced active AIP (Figure 3). Furthermore, all identified CA_C0082 homologs lack the conserved amino acids required for binding the essential zinc co-factor found in class I, II, and IV lanthipeptide synthetases and may therefore not carry out a cyclization reaction. However, class III lanthipeptide synthetases also lack the conserved zinc binding motif in their C-terminal domains and still function as cyclases [30,31], so while unlikely, a role in peptide cyclization cannot be ruled out yet. An exciting alternative hypothesis was that the remaining parts of the CA_C0082 homologs’ dehydratase domains may have retained their ability to carry out the phosphorylation of serine and threonine residues, thus acting as a general serine/threonine kinase, as demonstrated for Lacticin 481 synthetase [32]. Again, however, residues known to be important for this process [33,34] were either not or only partially conserved, and thus no final conclusions can be drawn as yet with regards to the molecular mechanism that defines the function of the CA_C0082 protein and its homologs in other species.

Association of peptide-based quorum sensing systems with lanthipeptide biosynthesis gene clusters is not uncommon in bacterial genomes, and in several species, these signaling systems have been confirmed to play a role in the regulation of lanthipeptide production, for instance in *Streptococcus pneumoniae* and *Staphylococcus epidermidis* [35,36]. Indeed, lanthipeptides themselves can act as signaling molecules, stimulating their own production in a quorum sensing-like manner [37]. Lanthipeptide biosynthesis gene clusters exist in most species harboring CA_C0082-homologous genes, although not *C. acetobutylicum*, and are in some instances located very close or immediately next to the respective *agrBD*-*agrCA*-‘CA_C0082′ cluster, for instance in *Clostridium hungatei* and *Clostridium cellulovorans*. Furthermore, full-length LanM genes found to be most similar to CA_C0082 (Appendix A) often appear to be part of larger lanthipeptide biosynthesis gene clusters that also include *agr* genes, for instance in *Clostridium saccharobutylicum* and some strains of *Bacillus paralicheniformis*. The observed co-localization lends support to the hypothesis that CA_C0082 homologs evolved from LanM proteins and took on new roles in regulatory pathways linked to cell-cell signaling.

In a previous study, Steiner et al. [10] observed sporulation defects in *C. acetobutylicum agr* quorum sensing mutants and speculated on how the AgrA response regulator may integrate with the regulatory cascade that controls endospore formation. An influence on Spo0A phosphorylation status was among the options considered. A highly revealing finding was therefore that mutations of CA_C0082 and *glgA* resulted in sporulation and solvent phenotypes very similar to those observed for *agr* mutants (Figure 5 and Figure 6). This suggested that a lack of granulose rather than direct regulatory control was primarily responsible for the observed reduction in sporulation. Indeed, the importance of granulose accumulation for spore formation became more pronounced as glucose concentrations were reduced (Figure 5).

Steiner et al. [10] also reported comparable final solvent concentrations for wild type and *agr* mutants, a finding confirmed here for all granulose-negative strains. Interestingly, however, the granulose-negative strains displayed increased acetone and butanol titers after 48 h (Figure 6), suggesting increased fermentation rates. A possible explanation for this phenomenon might be that following glucose uptake into the wild type cell, there is direct competition between the reactions of glucose catabolism and those leading to storage in the form of granulose, with the latter limiting glycolytic throughput. According to this hypothesis, the absence of granulose formation would allow glycolytic conversion and thus ABE fermentation to occur at a higher rate. Alternatively, higher 48 h mutant titers may have been a result of changes in metabolic regulation caused by granulose absence rather than direct pathway competition. This may explain the remarkably similar 48 h solvent titers observed for the wild type at glucose concentrations above 1%. The precise role of granulose in the overall physiology and metabolism of *C. acetobutylicum* merits further investigation, which was beyond the scope of this study. It seems clear, however, that a lack of granulose formation might be a positive feature of future engineered butanol production strains as it increases the rate of solvent production while having no effect on final levels. Shortened fermentation times would be of considerable benefit in industrial applications [38].

While the precise mechanism of CA_C0082 activity remains uncertain, our work suggests that it acts as a mediator of Agr quorum sensing, thereby controlling granulose production and, indirectly, sporulation. An interesting question is what advantage *C. acetobutylicum* might be able to gain by linking cell-cell communication, and thus, presumably, information on cell population density, with granulose synthesis. Granulose accumulation in response to population density may enable cells to build up internal glucose reserves well before external resources are depleted. This would make sugars inaccessible to competitors occupying the same niche while ensuring that individual cells have sufficient endogenous supplies to complete the process of endospore formation when external carbon and energy sources become limiting. In agreement with this hypothesis, we found a similar reduction in endospore formation for various mutant strains unable to generate granulose, which was more severe when glucose supply was limited (Figure 5). In *Bacillus subtilis,* initiation of endospore formation is triggered by nutrient starvation [39,40] and modulated in response to quorum sensing signals [41]. Information on cell density may be important in this context, as it can convey the degree of competition for already scarce nutrients [39] and allow a population to adjust sporulation levels accordingly. In *C. acetobutylicum* and other clostridial species, nutrient limitation has not been recognized as a general factor triggering sporulation [5], and the work presented here suggests that Agr quorum sensing does not contribute directly to the regulation of the sporulation cascade. Nevertheless, clostridial cells face similar constraints in that they must use their resources optimally to ensure maximum sporulation. A role for granulose as an endogenous source of carbon and energy for clostridial spore maturation was proposed decades ago [1,4]. Control of its synthesis by Agr quorum sensing may have evolved to ensure that bacteria can do so efficiently in their natural habitats by taking into account cell numbers and sugar availability. It is noteworthy that all species harboring *agr*-CA_C0082 clusters are either known to form granulose or to contain the required genes. Thus, the above considerations may also apply to the other species harboring this cluster, all of which are capable of forming spores apart from *Pseudobacteroides cellulosolvens* [42,43].

In addition to Agr quorum sensing and CA_C0082, several other regulatory factors are known to be important for granulose synthesis in solventogenic *Clostridium* species. These include Spo0A, Spo0E [7,44], and a recently discovered polyketide-based signaling system. The latter is of particular interest as a mutant unable to produce the polyketide Clostrienose was shown to exhibit a phenotype resembling that of *agr* and CA_C0082 mutants, displaying a granulose-negative phenotype and reduced spore formation [45]. However, the CA_C0082 protein is unlikely to mediate its effect via regulation of Clostrienose synthesis, as no chemical complementation of CA_C0082 mutants was observed when cross-streaked against the wild type. Future work on CA_C0082 might focus on elucidating its precise mechanism of action and contribution to the complex regulatory network that governs metabolism and cellular differentiation during and after the shift to solventogenesis. This could include transcriptomic analyses to identify the specific sets of genes affected by Agr quorum sensing, CA_C0082, and glycogen absence by looking at transcriptomic changes in the respective mutants. Another focus might be on the specific transcription factors that control the expression of *agr*, *ca_c0082,* and granulose genes themselves.

## Figures and Tables

**Figure 1 microorganisms-11-01460-f001:**
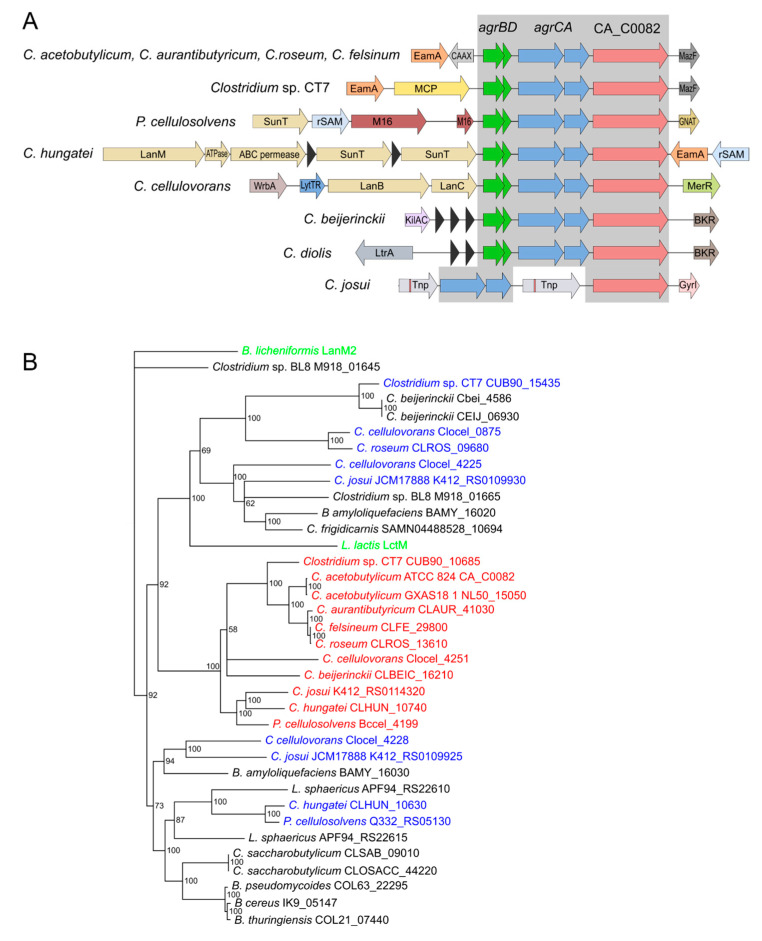
Conservation of *agrBDCA-*CA_C0082 gene clusters (**A**) and Bayesian MCMC phylogeny of CA_C0082 and LanM homologs (**B**). (**A**) The *clostridial agrBDCA-*CA_C0082 clusters are predicted to comprise three independent transcriptional units: *agrBD*, *agrCA,* and CA_C0082. Flanking regions are conserved in sequenced strains of the closely related species *C. acetobutylicum*, *C. roseum*, *C. aurantibutyricum*, and *C. felsinum*. Predicted gene products are abbreviated as follows: EamA: EamA-like transporter family protein; CAAX: CAAX-like membrane endopeptidase; MazF: mRNA-degrading endonuclease; MCP: methyl-accepting chemotaxis protein; SunT: ABC-type bacteriocin/lanthipeptide exporter; rSAM: radical SAM superfamily protein; M16: peptidase M16 superfamily; GNAT: N-acetyltransferase; LanM: LanM-like lanthipeptide synthetase; ATPase: ABC transporter ATPase component; ABC perm: ABC transporter permease component; SunT: ABC-type bacteriocin/lanthipeptide exporter; WrbA: multimeric flavodoxin; LytTR: LytTR family two-component transcriptional regulator; LanB: Lantibiotic dehydratase domain protein; LanC: LanC-like lanthipeptide synthetase; MerR: MerR family transcriptional regulator; KilAC: phage antirepressor protein KilAC domain; BKR: beta-ketoacyl-ACP reductase; LtrA: group II intron reverse transcriptase/maturase; Tnp: putative transposase, red lines indicating frameshifts; GyrI: GyrI-like small molecule binding domain. Black arrow heads: short open reading frames encoding proteins of unknown function. (**B**) The phylogenetic analysis of CA_C0082 homologs (red) included full-length LanM-type proteins present in the same strain (blue), established LanM proteins (green), and other LanM homologs to which CA_C0082 homologs were most similar, with LanM2 from *B. licheniformis* serving as an outgroup. CA_C0082 homologs form a distinct sub-cluster separate from any LanM proteins present in the same species. Only one representative locus tag is given for each species; identical proteins present in other strains are listed in Appendix A. Note: CA_C0082 homologs are found in only twelve of the currently sequenced *C. beijerinckii* strains (Appendix A), their sequence matching that of *C. diolis*.

**Figure 2 microorganisms-11-01460-f002:**
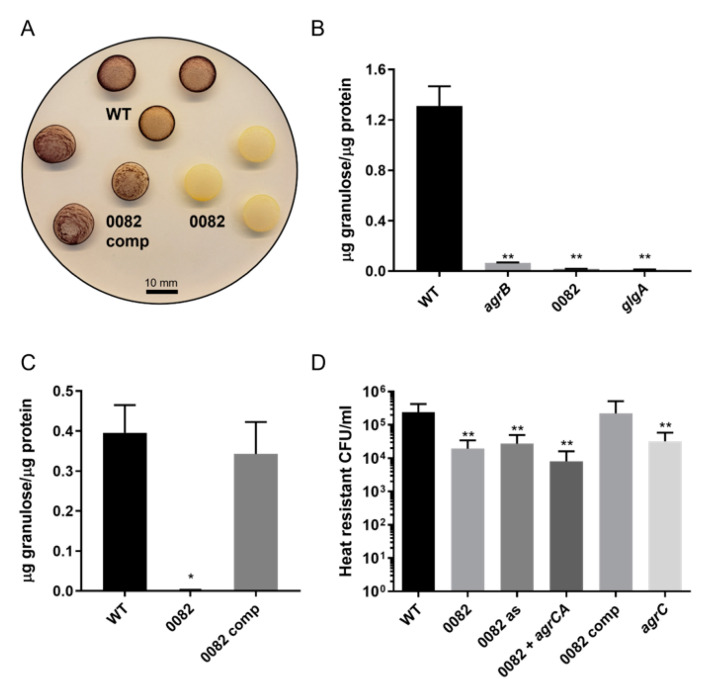
Granulose (**A**–**C**) and sporulation (**D**) phenotypes of *C. acetobutylicum* wild type, CA_C0082 mutant, and complemented CA_C0082 mutant. (**A**) Granulose accumulation was determined for *C. acetobutylicum* pMTL85141 (WT), ClosTron mutant *C. acetobutylicum* CA_C0082::CT*ermB*_s pMTL85141 (0082), and complemented ClosTron mutant *C. acetobutylicum* CA_C0082::CT*ermB*_s pMTL85141-CA_C0082 (0082 comp). Strains were grown as lawns on CBMS plates and subjected to iodine staining after 48 h. White cell lawns: absence of granulose or levels below the detection limit; brown cell lawns: granulose present. (**B**) Quantity of granulose produced per μg of protein after 48 h of growth in CBMS broth for *C. acetobutylicum* wild type (WT), *C. acetobutylicum* CA_C0082 (0082), *C. acetobutylicum agrB* (*agrB*), and *C. acetobutylicum glgA* (*glgA*) mutants. (**C**) Quantity of granulose produced per μg of protein after 48 h of growth in CBMS broth for *C. acetobutylicum* pMTL85141 (WT), *C. acetobutylicum* CA_C0082::CT*ermB*_s pMTL85141 (0082), and *C. acetobutylicum* CA_C0082::CT*ermB*_s pMTL85141-CA_C0082 (0082 comp). (**D**) For sporulation assays, the number of heat-resistant colony-forming units (CFU) was determined after 5 days of growth in CBMS broth. In addition to the strains shown in (**C**), sporulation assays included antisense ClosTron mutant *C. acetobutylicum* CA_C0082::CT*ermB*_as pMTL85141 (0082 as); strains *C. acetobutylicum agrC*::CT*ermB* pMTL85141 (*agrC*) and *C. acetobutylicum* CA_C0082::CT*ermB*_s pMTL85141-*agrCA* (0082 + *agrCA*) served as additional controls. Values shown are the means (±SD) of triplicate determinations for (**B**) and (**C**) and for three experiments each performed in technical triplicates for (**D**). Significance levels of the observed differences between mutants and wild type controls are indicated above the respective bars: *p* < 0.01, **; *p* = 0.01, *; no label: not significant.

**Figure 3 microorganisms-11-01460-f003:**
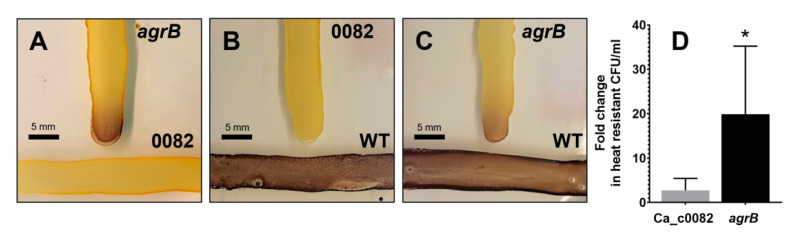
CA_C0082 mutants produce AIP but are unable to respond to it. AIP production was assessed using the cross-streak method. Strain CA_C0082::CT*ermB*_s (0082) was cross-streaked with (**A**) *agrB*::CT*ermB* (*agrB*) and (**B**) wild type *C. acetobutylicum* (WT). (**C**) Positive control showing *C. acetobutylicum* cross-streaked with *agrB*::CT*ermB*. Strains were streaked on CBMS plates and assayed for granulose accumulation after 48 h. (**D**) The number of viable heat-resistant endospores (heat-resistant CFU) formed by the CA_C0082 and *agrB* mutant strains was analyzed after 5 days of growth in CBMS broth in the presence of 20 mM AIP (R6T0) and compared to a DMSO control (fold change over control). The values shown are the means (±SD) of three independent experiments, each performed in technical triplicate. Significance levels of differences between AIP-supplemented cultures and the corresponding DMSO controls are indicated above the bars: *p* < 0.05, *; no label: not significant.

**Figure 4 microorganisms-11-01460-f004:**
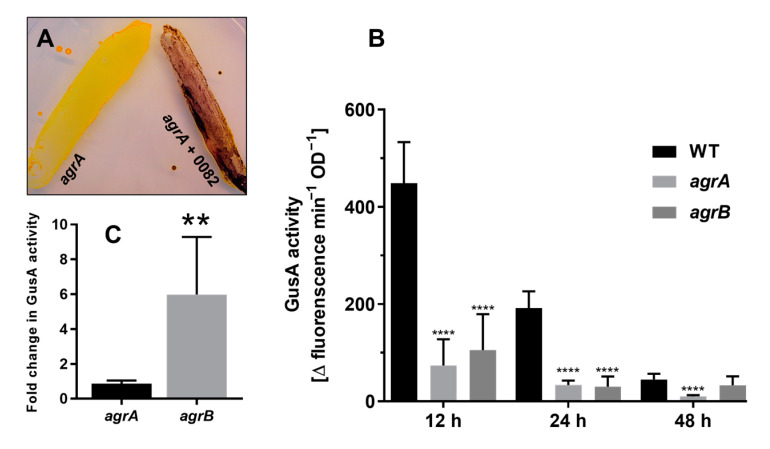
CA_C0082 expression is controlled by Agr quorum sensing. (**A**) Constitutive expression of CA_C0082 in an *agrA* mutant results in granulose production. Strains *agrA*::CT*ermB* pMTL85143_CA_C0082 (*agrA* + 0082), expressing CA_C0082 from a constitutive *fdx* promoter, and vector control *agrA*::CT*ermB* pMTL85143 (*agrA*) were assayed on CBMS plates for granulose accumulation after 48 h. (**B**) Expression of a transcriptional CA_C0082-*gusA* reporter (pMTL-JL1-P_CA_C0082_; with *gusA* under control of the CA_C0082 promoter) in wild type, *agrA,* and *agrB* mutant backgrounds after 12, 24, and 48 h of growth in CBMS, respectively, as indicated by GusA activity. (**C**) Relative expression of the CA_C0082-*gusA* reporter in *agrB* and *agrA* mutants after 12 h of growth in the presence and absence of 20 μM AIP (R6T0). Shown are the fold increases over the DSMO control. The values shown for each panel are the means (±SD) of three independent experiments, each performed in technical triplicate. Significance levels of differences between mutants and wild types are indicated above the bars: *p* < 0.0001, ****; *p* < 0.01, **; no label: not significant.

**Figure 5 microorganisms-11-01460-f005:**
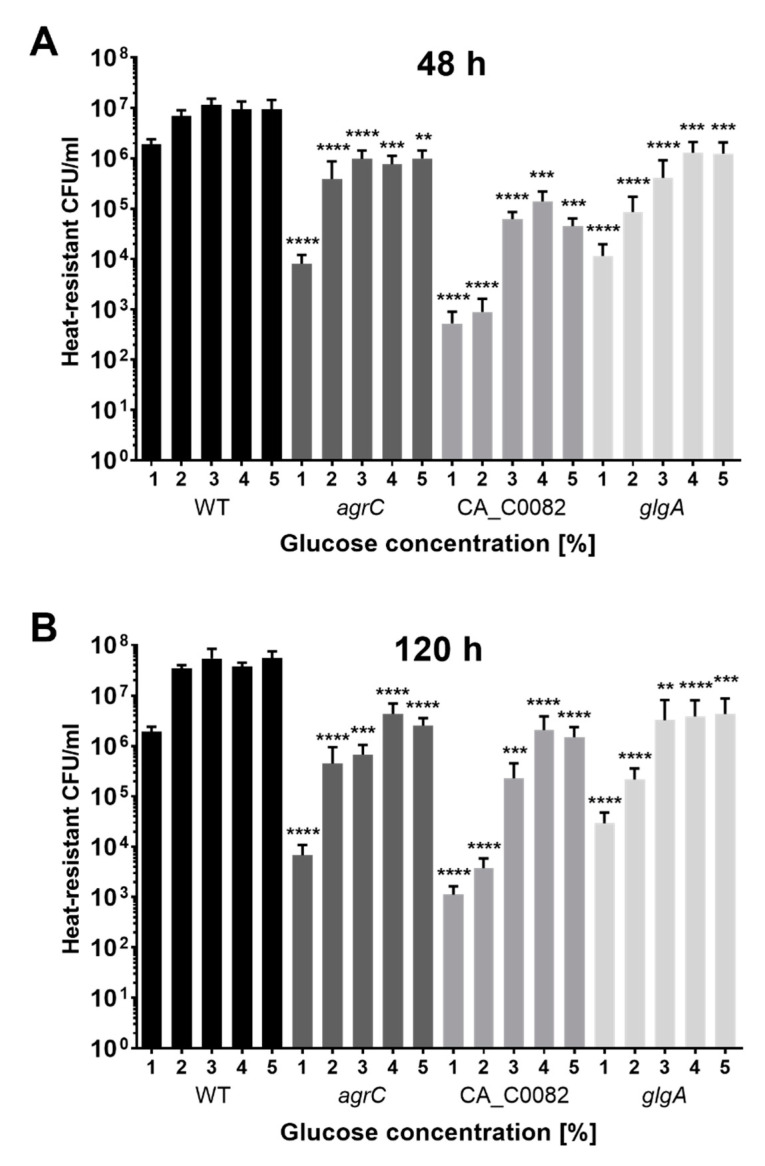
Mutants defective in CA_C0082, Agr quorum sensing, and the granulose biosynthesis pathway display similar sporulation profiles. *C. acetobutylicum* (WT, black) and mutant strains *agrC*::CT*ermB* (*agrC*, dark gray), CA_C0082::CT*ermB*_s (CA_C0082, medium gray), and Δ*glgA* (*glgA*, light gray) were grown in CBMS supplemented with 1–5% (*w*/*v*) glucose, and sporulation levels (heat-resistant CFU) were determined after (**A**) 48 h and (**B**) 120 h. The values shown are the means (±SD) of three independent experiments, each performed in technical triplicate. Significance levels of the observed differences between granulose-deficient mutants and the wild type control at each specific glucose concentration are indicated above the respective mutant strain bars: *p* < 0.0001, ****; *p* < 0.001, ***; *p* < 0.01, **; no label: not significant.

**Figure 6 microorganisms-11-01460-f006:**
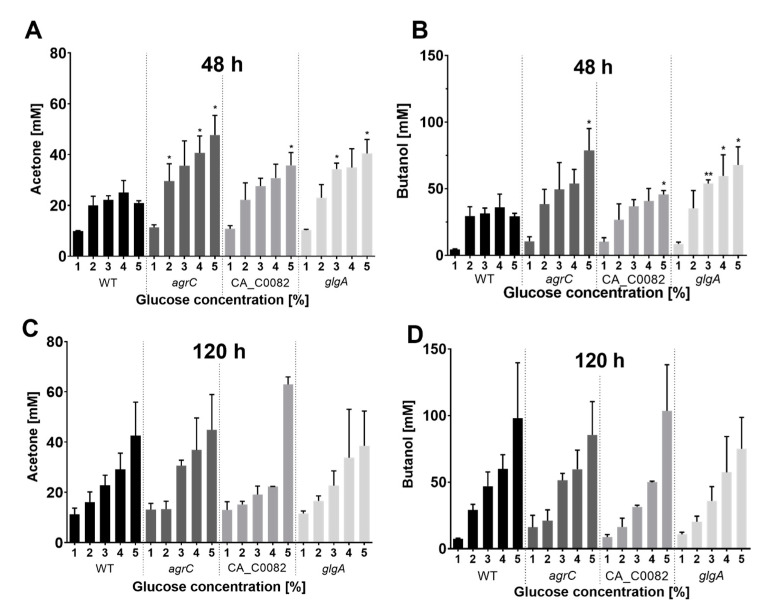
Mutants defective in CA_C0082, Agr quorum sensing, and the granulose biosynthesis pathway display similar solvent production profiles. *C. acetobutylicum* (WT, black) and mutant strains *agrC*::CT*ermB* (*agrC*, dark gray), CA_C0082::CT*ermB*_s (CA_C0082, medium gray), and Δ*glgA* (*glgA*, light gray) were grown in CBMS supplemented with 1–5% (*w*/*v*) glucose. Acetone (**A**,**C**) and butanol (**B**,**D**) concentrations in the culture supernatant were determined after 48 h (**A**,**B**) and 120 h (**C**,**D**). The values shown are the means (±SD) of three independent experiments, each performed in technical triplicate. Significance levels of the observed differences between granulose-deficient mutants and the wild type control at each specific glucose concentration are indicated above the respective mutant strain bars: *p* < 0.01, **; *p* < 0.05, *; no label: not significant.

**Table 1 microorganisms-11-01460-t001:** Bacterial strains used in this study.

Strain	Relevant Properties	Source/Reference
*E. coli* Top10	F-*mcrA* Δ(*mrr-hsdRMS-mcrBC*) Φ80*lacZ*ΔM15 Δ*lacX*74 *recA*1 *araD*139 Δ(*ara leu*)7697 *galU galK rpsL* (StrR) endA1 nupG	Invitrogen
*E. coli* Top 10 pAN2	*E. coli* Top 10 with methylation plasmid pAN2 containing the ϕ3TI methyltransferase	[18]
*C. acetobutylicum* ATCC 824	*C. acetobutylicum* ATCC 824 wild type	Prof. Hubert Bahl, University of Rostock (COSMIC-strain)
*C. acetobutylicum* CA_C0082::CT*ermB*_s	*C. acetobutylicum* ATCC 824 CA_C0082 ClosTron mutant; ClosTron inserted in “sense” orientation between base pairs 1119/1120	This study
*C. acetobutylicum* CA_C0082::CT*ermB*_as	*C. acetobutylicum* ATCC 824 CA_C0082 ClosTron mutant; ClosTron inserted in “antisense” orientation between base pairs 538/539	This study
*C. acetobutylicum agrA*::CT*ermB*	*C. acetobutylicum* ATCC 824 *agrC* ClosTron mutant; unable to mediate a quorum sensing response	[10]
*C. acetobutylicum agrB*::CT*ermB*	*C. acetobutylicum* ATCC 824 *agrB* ClosTron mutant; unable to produce AIP (quorum sensing peptide)	[10]
*C. acetobutylicum agrC*::CT*ermB*	*C. acetobutylicum* ATCC 824 *agrC* ClosTron mutant; unable to sense AIP (quorum sensing peptide)	[10]
*C. acetobutylicum* Δ*glgA*	*C. acetobutylicum* ATCC 824 glycogen synthase (*glgA*) in-frame deletion mutant	[19]
*C. acetobutylicum* pMTL85141	ATCC 824 wild type with empty pMTL85141 vector	This study
*C. acetobutylicum* CA_C0082::CT*ermB*_s pMTL85141	CA_C0082 mutant with empty pMTL85141 vector	This study
*C. acetobutylicum* CA_C0082::CT*ermB*_s pMTL85141-CA_C0082	CA_C0082 mutant with pMTL85141-CA_C0082 complementation vector (native CA_C0082 promoter)	This study
*C. acetobutylicum* CA_C0082::CT*ermB*_s pMTL85141-*agrCA*	CA_C0082 mutant with pMTL8514-*agrCA* complementation vector	This study
*C. acetobutylicum agrA*::CT*ermB* pMTL85143	*agrA* mutant with empty pMTL85143 (ferredoxin promoter)	This study
*C. acetobutylicum agrA*::CT*ermB* pMTL85143-CA_C0082	*agrA* mutant with pMTL85143-CA_C0082 vector (ferredoxin promoter)	This study
*C. acetobutylicum* pMTL*-*JL1-P_CA_C0082_	ATCC 824 wild type expressing CA_C008*2-gusA* transcriptional reporter fusion	This study
*C. acetobutylicum agrA*::CT*ermB* pMTL-JL1-P_CA_C0082_	*agrA* mutant expressing CA_C008*2-gusA* transcriptional reporter fusion	This study
*C. acetobutylicum agrB*::CT*ermB* pMTL-JL1-P_CA_C0082_	*agrB* mutant expressing CA_C008*2-gusA* transcriptional reporter fusion	This study

**Table 2 microorganisms-11-01460-t002:** Plasmids used in this study.

Plasmid	Relevant Properties	Source
pAN2	Plasmid containing ϕ3TI methyltransferase	[18]
pMTL007C-E2::CA_C0082-538|539A	ClosTron plasmid retargeted to CA_C0082 ^1^	This study
pMTL007C-E2::CA_C0082-1119|1120S	ClosTron plasmid retargeted to CA_C0082 ^1^	This study
pMTL85141	*Clostridium* modular plasmid containing a *catP*resistance marker	[20]
pMTL85143	pMTL85141 with *C. sporogenes* ferredoxin promoter upstream of a multiple cloning site	Ref. [11], Dr Ying Zhang, Univ. of Nottingham
pMTL85141-CA_C0082	pMTL85141 containing CA_C0082 coding region and 88 bp non-coding upstream promoter region	This study
pMTL85141-*agrCA*	pMTL85141 containing the *C. acetobutylicum agrCA* operon under control of its native promoter	[10]
pMTL85143-CA_C0082	pMTL85143 containing the CA_C0082 coding region under control of consitutive *C. sporogenes* ferredoxin promoter	This study
pMTL-JL1	pMTL85143 derivative conatining the *Streptococcus agalactiae gusA* gene under control of the *C. sporogenes* ferredoxin promoter	Dr Jessica Locke, Univ. of Nottingham
pMTL-JL1-P_CA_C0082_	CA_C0082 β-glucuronidase reporter; pMTL-JL1 derivative containing a transcriptional fusion of the *C. acetobutylicum* CA_C0082 promoter region and *Streptococcus agalactiae gusA*	This study

^1^ Numbers following the gene name indicate the predicted insertion site of the encoded ClosTron derivative, with S and A denoting sense and anti-sense orientation, respectively.

## Data Availability

All data generated or analyzed during this study are included in this published article and its Appendix A.

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
