# Peer review of "The Lanthipeptide Synthetase-like Protein CA_C0082 Is an Effector of Agr Quorum Sensing in *Clostridium acetobutylicum"

_microorganisms, 2023, doi:10.3390/microorganisms11061460_

Round 1

Reviewer 1 Report

In the manuscript „The lanthipeptide synthetase-like protein CA_C0082 is an effector of Agr quorum sensing in Clostridium acetobutylicum“, Humphreys, Bean, and co-workers report a LanM-like protein encoded in proximity to the agr quorum sensing operon in a Clostridium acetobutylicum strain. Their data shows clearly that CA_C0082 is involved in granulose formation and is regulated by the Agr quorum sensing system. Despite its homology to LanM lanthipeptide synthetases, CA_C0082 is however not involved in the formation of the autoinducing peptide derived from AgrD.

This manuscript is very well written and the experimental design is convincing and contains all the necessary controls to support the conclusions drawn by the authors. The scholarly presentation is also excellent. Taken together, I would greatly recommend the publication of this manuscript!

I only have a minor comment for the authors to consider:

- The comparison of CA_C0082 to known LanM lanthipeptide synthetases is mainly focused on the C-terminal cyclization domains. However, I miss the comparison of the N-terminal dehydratase domains, as these could be especially relevant for the proposed role of CA_C0082 in signaling. Concretely, lanthipeptide precursor processing by LanMs begins with the ATP-dependent phosphorylation of Ser/Thr sidechains. Hence, it could be feasible that CA_C0082, if it still contains the corresponding catalytic residues, might have evolved to carry out the phosphorylation of proteins involved in signaling cascades.

Therefore, I would suggest the authors to look more closely at the N-terminal region of CA_C0082 and compare it again to LanM dehydratase domains. In general, I think that an alignment of CA_C0082-type proteins with known LanM enzymes should be included in the main text instead of just being places in the supporting information!

It could furthermore be interesting to do a structure prediction for the cyclase domain of CA_C0082 with known structures of LanM cyclase domains and compare the predicted active site with the ones of known structures.

Author Response

Response 1:

We are grateful for the reviewer’s supportive comments and suggestions, which were addressed as follows:

The suggestion that conserved residues within the N-terminal dehydratase domains might still enable phosphorylation of serine/threonine side chains is absolutely great and admittedly one that we did not think of, given the truncations found within this domain for our identified CA_C0082 homologs. We had also been unaware of the work done by You et al. (2009) and others on e.g. Lacticin 481 synthetase, demonstrating its ability to act as a general serine/threonine kinase.

Following up on this exciting suggestion, we took a closer at the dehydratase domain of CA_C0082 and its close homologs, but to our disappointment conserved residues of importance for the phosphorylation step were either not or only partially conserved, as shown in the alignment now added as Supplementary Fig. S3. This information has also been added to the main text (see below). Nevertheless, there is some degree of sequence conservation within the regions where most of these residues are located and we have added this point to the discussion section.

Regarding showing an overall alignment of CA_C0082 homologs and established LanMs in the main text, having undertaken this alignment we realized that due to the large size of the proteins shown, this would cover several pages even when set to font size 6. Because of this, and in the absence of intriguing new leads coming from it, we decided to provide the alignment of complete sequences as a separate supplementary figure instead.  We do however refer to the drawn conclusions in the main text (section 3.2, line 260 onwards).

This now reads:

“However, sequence comparison of the different CA_C0082 protein homologs with established lanthipeptide synthetases revealed that their cyclase domains lacked conserved motives and amino acid residues required for binding of an active site zinc ion (Supplementary Fig. S2) [28]. Similarly, conserved residues in the dehydratase domain with established roles in the phosphorylation and elimination steps were also lacking (Supplementary Fig. S3). Indeed, while the CA_C0082 protein and its homologs were found to be most similar to LanM-type lanthipeptide synthetases, they were somewhat smaller, consisting of 670 to 728 amino acids rather than the typically observed 900-1200 residues [29], mainly due to absence of parts of the dehydratase domain (Supplementary Fig. S4)."

We have also amended our discussion to include this point from line 537 onward:

“An exiting alternative hypothesis was that the remaining parts of the CA_C0082 homo-logs’ dehydratase domains may have retained their ability to carry out the phosphorylation of serine and threonine residues, thus acting as a general serine/threonine kinase as demonstrated for Lacticin 481 synthetase [32]. Again, however, residues known to be important for this process were either not or only partially conserved and thus no final conclusions can be drawn as yet with regards to the molecular mechanism that defines the function of the CA_C0082 protein and its homologs in other species.”

Given the observed lack of conserved residues we did not model the structures of either cyclase or dehydratase domains, but completely agree that this will be very interesting to follow up in in our future investigations.

Reviewer 2 Report

The physiological role and mode of action of the encoded LanM-like proteins remains uncertain as they lack conserved catalytic residues. In this manuscript, Winzer and coworkers presented for the industrial organism Clostridium acetobutylicum that the LamM-like protein CA_C0082 is not required to produce active AgrD-derived signaling peptide, but nevertheless acts as an effector of Agr quorum sensing. The results showed that expression of CA_C0082 was controlled by the Agr system and is a prerequisite for granulose formation. Accumulation of granulose, in turn, was shown to be required for maximal spore formation but also to reduce early solvent formation. CA_C0082 and its putative homologs appear to be closely associated with Agr systems predicted to employ signaling peptides with six-membered ring structures and may represent a new subfamily of LanM-like proteins. In summary, this manuscript was well-written, concise, and the experiments were carefully designed and performed. The significance and novelty were clearly stated that this is the first time bacterial Agr signaling has been described in detail. This finding also motivated the study of CA_C0082 in the context of Agr-dependent quorum sensing. More specifically, the aim of the work presented here was to establish whether the CA_C0082 gene product was required for generation of the Agr signaling peptide or contributed in other ways to the expression of Agr-dependent phenotypes. Only some minor changes need to be addressed before final publication.

Comments:

1)    Throughout the text, “signalling” should be spelled as “signaling”.

2)    In section 2.4 and 2.5 for the Methods and Materials, the format of restriction enzymes should change. It should be “NotI” but not “NotI”, “NheI” but not “NheI’, etc. Please change the formats for all the restriction enzymes that mentioned.

3)    In line 172, should be “synthesized”.

4)    In Figure 2, for part A, are wild type colonies should look similar to 0082 comp colonies? And why are these two types similar? Should explain in detail in the Figure descriptions as well. What is the scale for part A? Should include a “scale meter” in the Figure. Similarly, “scales” should be put into Figure 3, part A and Figure 4, part A.

5)    In Figure 3, why the error bar for part D agrB strain was that large? The authors should explain why they got this huge error bar.

6)    Also, in Figure 4, part C agrB, the error bar was large. Please explain.

7)    In line 439, the sentence “For this, use was made of the…” is not clear to me. I am not sure what ideas the authors were trying to convey. Please rephrase.

8)    The authors mentioned that for future work on CA_C0082, precise mechanism of action and contribution to the complex regulatory network will be studied. Will the authors also going to dive deeper into transcriptomics or transcription factors that affect the regulatory network?

Overall English is great, only minor changes are needed.

Author Response

Reviewer 2:

We are grateful for the reviewer’s helpful and constructive comments, which we have addressed as follows:

1) Throughout the text, “signalling” should be spelled as “signaling”.

Response 1:

We wrote our manuscript in British English hence the use of “signalling”. We have now changed to “signaling’ as requested by the reviewer.

2) In section 2.4 and 2.5 for the Methods and Materials, the format of restriction enzymes should change. It should be “NotI” but not “NotI”, “NheI” but not “NheI’, etc. Please change the formats for all the restriction enzymes that mentioned.

Response 2:

The format change requested by the reviewer does not comply with current nomenclature rules, which changed some time ago (Roberts et al. (2003) A nomenclature for restriction enzymes, DNA methyltransferases, homing endonucleases and their genes. Nucleic Acids Res. 31(7): 1805–1812). After seeking guidance from the editorial office, we were instructed to leave the format unchanged (not italicised).

3)  In line 172, should be “synthesized”.

Response 3:

We have changed to “synthesized’ as requested and also “characterised” to “characterized”, “organised” to “organized” etc.

4)    In Figure 2, for part A, are wild type colonies should look similar to 0082 comp colonies? And why are these two types similar? Should explain in detail in the Figure descriptions as well. What is the scale for part A? Should include a “scale meter” in the Figure. Similarly, “scales” should be put into Figure 3, part A and Figure 4, part A.

Response 4:

Figure 2A, relating to colonies:

We should have made it clearer that this figure shows cells lawns, not colonies, that formed after depositing suspensions of the respective strains on a CBMS agar surface. We have now added this information to the figure legend and also expanded our description of the process in the Methods section 2.6. As requested, we explain similarities between the formed lawns in the legend, (i.e. that all dark stained lawns are positive for granulose and those white stained are not.

With respect to similarities other than granulose containing cells being stained by iodine, the referee presumably wonders about the surface structure of the lawns, which is visible rougher for the complemented strain. Most likely this is a consequence of the cells in the deposited suspensions aggregating differently during the drying process. This is interesting but we did not follow up on it, as the point of the experiment was to show that loss of granulose formation was indeed linked to loss of the CA_C0082 gene. As the reviewer will be aware, genetic complementation can be accompanied with other effects linked e.g. to increased gene copy number of the complementing gene and/or changes to its expression profile.

With respect to scale bars: these have been added to all Figure 2A, Figure 3A and Figure 4A, as requested.

5)    In Figure 3, why the error bar for part D agrB strain was that large? The authors should explain why they got this huge error bar.

Response 5:

We cannot be absolutely certain why the error was larger for this particular experiment but please note that this particular graph relates to endospore numbers, i.e. a stage reached after a long and complex developmental cycle. We believe that for experiments such as this, small differences in the starting conditions for the biological replicates are prone to result in more marked differences later on. This also applies to the timing of transient processes such as granulose synthesis and degradation, and those depending on signaling, as shown in Figure 4C, creating larger standard deviations. For endospore quantification and more difficult-to-grow organisms, error bars can on occasion cover ranges that deviate more substantially from the mean, but due to the high numbers observed this is often shown in the form of log plots were such differences are less obvious than in the figures discussed here, where we show ratios.

As for all such experiments there is “noise” in the data but the crucial point is that the changes we are showing here are nevertheless statistically significant, with p values of 0.05 and below. 

6)    Also, in Figure 4, part C agrB, the error bar was large. Please explain.

Response 6:

Please see the explanation given above for Figure 3.

7)    In line 439, the sentence “For this, use was made of the…” is not clear to me. I am not sure what ideas the authors were trying to convey. Please rephrase.

Response 7:

Apologies for being unclear. At this point in text, we simply wanted to reiterate that we had access to a previously generated granulose-negative mutant which we used for our experiment. As requested, we have rephrased the sentence which now reads “A previously generated glgA (granulose synthase) mutant incapable of synthesizing granulose [22] was employed for this purpose and served as a comparison. [22].”

8)    The authors mentioned that for future work on CA_C0082, precise mechanism of action and contribution to the complex regulatory network will be studied. Will the authors also going to dive deeper into transcriptomics or transcription factors that affect the regulatory network?

Response 8:

Yes, this is indeed the case. For instance, we intend to investigate the sets of genes affected by Agr quorum sensing, CA_C0082 and glycogen absence by looking at transcriptomic changes in the respective mutants. Another focus will be on the specific transcription factors that control the expression of agr, ca_c0082 and granulose genes themselves.

We have added this information in the last paragraph of the Discussion section.